# WorldWeaver: Generating Long-Horizon Video Worlds via Rich Perception

**Zhiheng Liu[1,2], Xueqing Deng[2], Shoufa Chen[1], Angtian Wang[2], Qiushan Guo[2], Mingfei Han[2], Zeyue Xue[1], Mengzhao Chen[1], Ping Luo[1†], Linjie Yang[2†]**

[1]The University of Hong Kong, [2]ByteDance Seed

## Abstract

Generative video modeling has made significant strides, yet ensuring structural and temporal consistency over long sequences remains a challenge. Current methods predominantly rely on RGB signals, leading to accumulated errors in object structure and motion over extended durations. To address these issues, we introduce WorldWeaver, a robust framework for long video generation that jointly models RGB frames and perceptual conditions within a unified long-horizon modeling scheme. Our training framework offers three key advantages. First, by jointly predicting perceptual conditions and color information from a unified representation, it significantly enhances temporal consistency and motion dynamics. Second, by leveraging depth cues, which we observe to be more resistant to drift than RGB, we construct a memory bank that preserves clearer contextual information, improving quality in long-horizon video generation. Third, we employ segmented noise scheduling for training prediction groups, which further mitigates drift and reduces computational cost. Extensive experiments on both diffusion- and rectified flow-based models demonstrate the effectiveness of WorldWeaver in reducing temporal drift and improving the fidelity of generated videos. Page could be found here.

## 1   Introduction

Long-horizon video prediction [10, 76, 11, 77, 17, 45, 46] is a fundamental challenge for world modeling, as it requires models to internalize and reproduce the causal laws that govern scene dynamics over time. Although generative video modeling [65, 34, 34, 1, 48, 47, 49], has made rapid progress and shows impressive performance on short sequences less than 10 seconds, preserving realism and coherence over longer durations remains a significant challenge. Current approaches often struggle to depict correct motion and shape in long-horizon predictions, resulting in severe degradation including objects warp, motions drift, and violations of physical constraints as time progresses. These challenges highlight the difficulty of preserving both dynamics and temporal consistency in long video generation, which is critical for structured and predictive understanding of the world.

Most generative video models nowadays rely exclusively on RGB information as training signals, optimizing pixel-level reconstruction objectives. This causes them to favor color and texture over structural and dynamical attributes, such as motion trajectories, object geometry, and scale relationships [9, 40, 18]. In other words, training solely on RGB information can lead to the aforementioned instabilities. Such instabilities can be magnified over long sequences as small frame-wise errors accumulate and compound, resulting in pronounced temporal drift and structural degradation.

39th Conference on Neural Information Processing Systems (NeurIPS 2025).

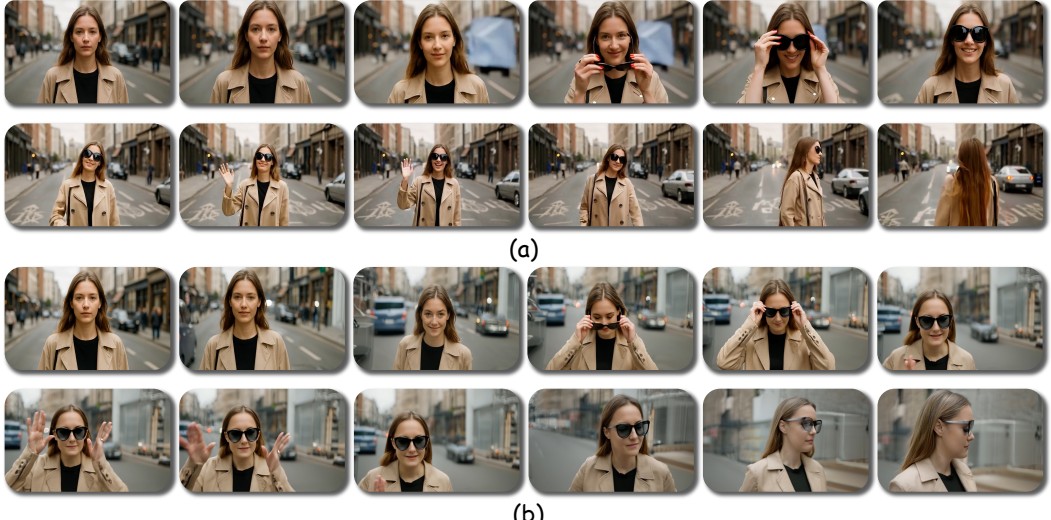

(a)

(b)

Prompt: A woman walks down the street, smiling as she puts on her sunglasses. She stops to wave hello, then turns around and walks away.

Figure 1: **WorldWeaver vs. existing approach on Long-Horizon Video Generation.** Compared to other methods (b), WorldWeaver (a) achieves superior temporal consistency and motion quality in long-horizon video generation.

To address these issues, as shown in Fig. 1, we introduce WorldWeaver, a robust framework for accurate long-horizon video prediction that seamlessly integrates joint modeling of RGB frames and perceptual signals. Our approach employs a perceptual-conditioned next prediction scheme to achieve precise and stable long-video generation. Our method encourages the model to capture not just color, but also structural and motion-related cues, thereby improving the robustness of the generated outputs. To achieve this, we incorporate multiple perceptual signals-such as video depth and optical flow-into a joint training framework. Extensive experiments confirm the effectiveness of using these auxiliary signals. Inspired by the observation that depth is more stable than RGB under temporal drift, we introduce a memory bank that stores reliable historical information from previous frames to enhance temporal consistency. This allows our model to maintain clearer long-term context and effectively mitigate the accumulation of drift in RGB predictions. In contrast, prior methods [10, 60, 77, 24] often inject substantial noise into historical information to prevent drift, which diminishes the utility of long-range context. Additionally, we adopt a segmented noise scheduling strategyassigning different noise levels to separate segments during training and constructing delayed denoising prediction groups during inferenceto further mitigate drift and reduce computational cost.

To evaluate our framework, we verify its effectiveness on both diffusion-based (CogVideoX-1.5B) and rectified flow-based (Wan2.1-1.3B) models on a robotic-arm manipulation dataset and a general-purpose video dataset. Extensive experiments demonstrate that our method substantially improves the stability and consistency of long-horizon video generation, effectively reducing temporal drift and structural distortions across diverse scenarios.

Our contributions can be summarized as follow:

- Systematically exploring the role of image-based perceptual condition, such as depth and optical flow, in enhancing long-horizon video generation as auxiliary signals.

- Proposing a unified framework that integrates perceptual conditioning and memory mechanisms for robust long-horizon video prediction.

- Extensive validation across different generative models and datasets, including both general-purpose and robotic manipulation domains, highlighting the potential of our approach as a foundation for scalable world models.

## 2  Related Works

**Long video generation.**  With recent advances in video diffusion models [5, 34, 63, 38, 12], long video synthesis has garnered increasing attention for applications such as storytelling and simulations. Existing approaches span GAN-based methods [22, 41], training-free techniques [52, 64], distributed inference [61], autoregressive frameworks [74], and hierarchical strategies [31, 73]. Among these, FreeNoise [53] extends diffusion models by rescheduling noise without fine-tuning; StreamingT2V [32] integrates autoregressive modeling with memory modules for improved coherence; and TTT [13] adapts models at test time using specialized layers. LCT [24] enhances pre-trained models with 3D RoPE and asynchronous noise scheduling to produce scene-consistent multi-shot videos, while FAR [23] employs frame-level autoregressive modeling with FlexRoPE and context strategies for efficient long video generation. However, most existing methods rely solely on RGB information from historical frames, leading to cumulative structural and motion errors. In contrast, our approach incorporates perceptual signals—such as depth and optical flow—from historical frames to more effectively guide future frame generation.

**Generative model for world simulation.**  World models, originating from foundational work [26], have evolved from RNN-based latent dynamics [27–29] to generative approaches such as diffusion-based [2, 15, 62, 19, 1] and autoregressive models [6, 44, 56, 39, 1]. Unlike agent-centric prediction, physical-world simulation emphasizes physics-informed realism for real-world applications. Early efforts aim to enhance simulator outputs: RL-CycleGAN [54] refines rendered images while aligning utility through Q-values, and RetinaGAN [33] preserves object features in simulated scenes for reinforcement learning. Recent diffusion [65, 57, 4] and autoregressive diffusion [10, 60, 74] frameworks surpass GANs by producing higher-fidelity, more temporally consistent, and realistic dynamic outputs. These advances enable video generation models to serve as learnable simulators, powering physics-grounded simulation in applications such as game environments [6, 62, 39, 7, 75, 14, 71], autonomous driving [35, 37, 51, 69, 20, 21], and robotic control [3, 16, 8, 30, 25, 78]. However, current video generation frameworks often prioritize color and texture over robust modeling of dynamics and spatial relationships [40, 9]. During inference, they frequently struggle to incorporate real-world knowledge beyond historical RGB data, limiting their ability to simulate complex interactions accurately.

## 3  Methods

In this section, we first introduce the preliminary and background on video diffusion models (Sec. 3.1). To enhance structure and motion modeling capabilities in video generation, we propose to jointly learn perceptual cues, such as depth and optical flow from video data that captures rich spatial structures and temporal dynamics (Sec. 3.2). We further propose a perception-conditioned long-horizon generation framework, which leverages a perceptual memory bank and group-wise noise scheduling to maintain robust context and mitigate drift in long video prediction (Sec. 3.3).

### 3.1  Preliminary

Video diffusion models offer a powerful framework for generating high-quality videos from conditional inputs.  These models generate coherent video sequences by learning to reverse a noise-injection process, progressively refining random noise into structured outputs.

We adopt flow matching [43] which offers a deterministic alternative to stochastic diffusion processes. Flow matching constructs a continuous trajectory from a noise distribution to the target video distribution, conditioned on an input $y$, representing the conditioning signal (e.g., a text prompt). Formally, given a clean video sample $x_1 \in \mathbb{R}^{T \times C \times H \times W}$, a noise sample $x_0 \sim \mathcal{N}(0, I)$, and the condition $y$, the interpolation at timestep $t \in [0, 1]$ is: $x_t = tx_1 + (1 - t)x_0$. The model learns the conditional velocity of this trajectory, defined as: $v_t = \frac{dx_t}{dt} = x_1 - x_0$. The training objective minimizes the discrepancy between the predicted velocity $u(x_t, y, t; \theta)$ and the true velocity:

$$\mathcal{L}(\theta) = \mathbb{E}_{x_1, x_0 \sim \mathcal{N}(0,I), y, t \in [0,1]} \left[ \|u(x_t, y, t; \theta) - v_t\|_2^2 \right], \qquad (1)$$

where $\theta$ denotes the model parameters and $y$ is the conditioning input.  During inference, an ordinary differential equation (ODE) solver integrates the velocity field to generate videos from

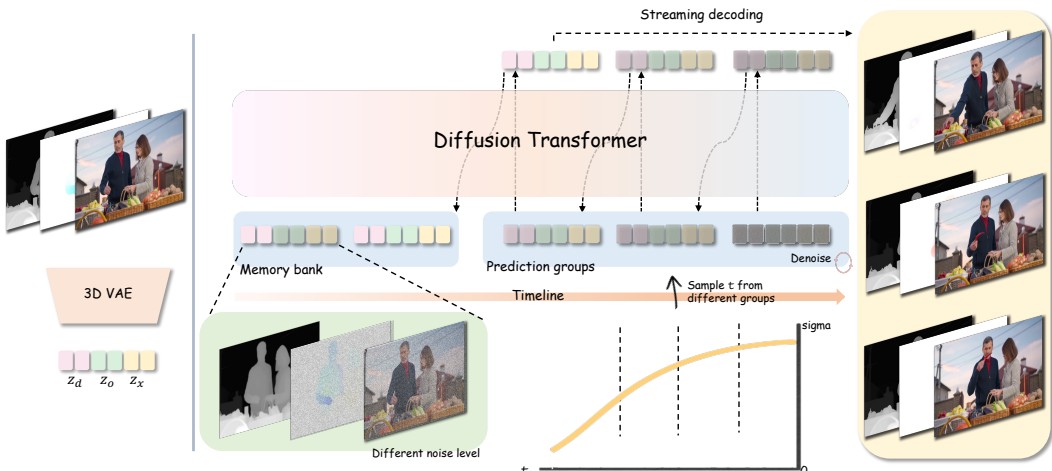

Figure 2: **Framework of WorldWeaver.** Given an input video, RGB, depth, and optical flow signals are encoded into a joint latent representation via a 3D VAE. The latents are split into a memory bank and prediction groups for the Diffusion Transformer. The memory bank stores historical frames and is excluded from loss computation; short-term memory retains a few fully denoised frames for fine details, while long-term memory keeps depth cues noise-free and adds low-level noise to RGB information. During training, prediction groups are assigned different noise levels according to the noise scheduler curve, aligned with the noise scheduling used during inference.

noise, guided by $y$. To mitigate the computational complexity of high-dimensional video data, models employ a pretrained variational autoencoder (VAE) to compress video sequences into a latent space. Given a video sequence $X \in \mathbb{R}^{F \times C \times H \times W}$, where $F$, $C$, $H$, and $W$ represent the number of frames, channels, height, and width, respectively, the VAE maps it to a latent representation $z_x \in \mathbb{R}^{f \times c \times h \times w}$. This compression reduces the temporal and spatial dimensions while preserving essential features, facilitating efficient optimization in the latent space. Diffusion Transformer serves as the primary network architecture, leveraging attention mechanisms to model long-range spatio-temporal dependencies.

## 3.2 Enhancing Video Generation with Perceptual Conditions

Recent studies [40, 9] reveal a critical limitation in current diffusion-based video generation approaches: the learning is prioritized towards color information, leading to a neglect of essential attributes such as motion, relative size, shape, and spatial relationships. This bias hampers the ability of models to capture the dynamic and structural complexities of video data, particularly in next-frame prediction frameworks for long video generation. Here, distortions in shape, size, or motion in individual frames can trigger rapid error accumulation, leading to incoherent sequences. To address this, we introduce a unified modeling framework that integrates RGB data with perceptual conditions, specifically video depth and optical flow. Additionally, we evaluate the impact of various perceptual conditions, including video segmentation, with findings detailed in Sec. 4.3.

We first describe the process of preparing depth and optical flow data for our training videos. For depth estimation, we employ Video Depth Anything [68] to generate initial depth maps for each frame of the input video. Then we use DepthAnythingV2 [70] to compute relative depth across frames through least-squares optimization. The resulting depth sequences are normalized to form $D \in \mathbb{R}^{F \times 1 \times \tilde{H} \times W}$. To match the dimensionality of the VAE input, we repeat the single channel three times, giving $D \in \mathbb{R}^{F \times 3 \times H \times W}$. This depth representation is then compressed into a latent space using a variational autoencoder (VAE): $z_d = \mathcal{E}(D)$, where $\mathcal{E}$ denotes the VAE encoder.

Next, we employ SEA-RAFT [66] to extract optical flow between consecutive frames of the input video, producing a displacement field $O \in \mathbb{R}^{(F-1) \times 2 \times H \times W}$, where $O(u, v)$ represents the displacement of the pixels between frames. To encode this into a format compatible with our framework, we convert $O$ into an RGB representation by computing the magnitude and direction of motion of each pixel, defined as $m = \min\left\{1, \frac{\sqrt{u^2+v^2}}{\sigma\sqrt{H^2+W^2}}\right\}$ and $\alpha = \arctan 2(v, u)$, with $\sigma = 0.15$,

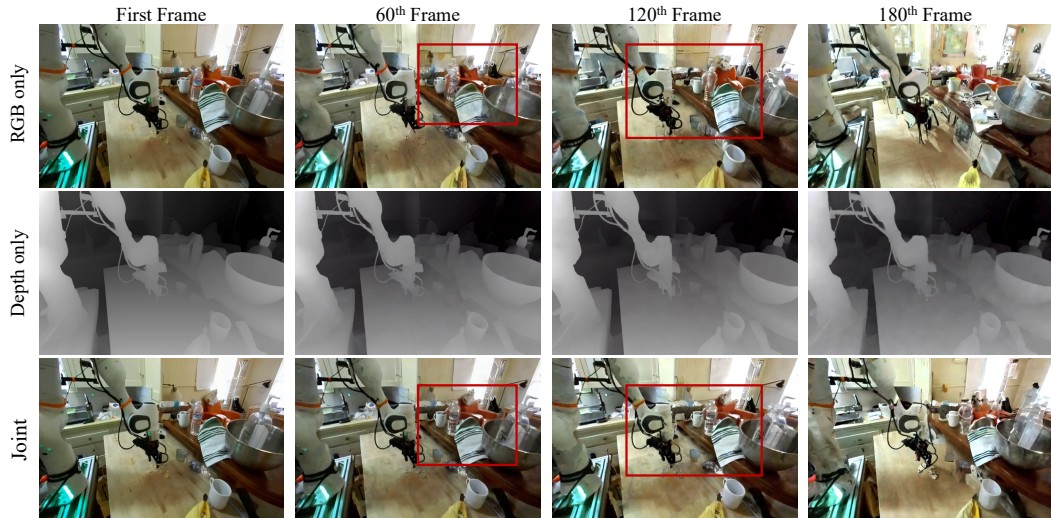

Figure 3: **Visualization of drift resistance.** We construct video data by repeating a single image during training and include "static" in the prompt. When the model is set to output only RGB, severe structural degradation and color distortion occur as the number of frames increases. In contrast, when outputting only depth, the structure remains well preserved. Jointly outputting both RGB and depth significantly alleviates visual drift. See experimental details in supplementary materials.

following VideoJAM [9]. Here, $m$ determines pixel opacity, and $\alpha$ assigns a color based on the motion direction. To align with the VAE input, we pad $O$ with a zero frame to match the temporal dimension of the input video, forming $O \in \mathbb{R}^{F \times 3 \times H \times W}$. This flow representation is then compressed into a latent space using the VAE encoder: $z_o = \mathcal{E}(O)$. Note that we use "synthetic" depth and optical flow data from open source predictive models, since obtaining ground-truth depth/flow is not possible with large-scale web videos.

To integrate appearance, depth, and optical flow into a unified latent representation, we extend the input and output layers of the pretrained model to accommodate the additional channels for depth and flow. For the RGB input channels, we initialize the model branch from the corresponding pretrained weights to reduce training cost, while the channels corresponding to depth and flow are randomly initialized to allow the model to adapt to these modalities during fine-tuning. As shown on the left of Fig. 2, after encoding, the latent representations of the video, depth, and optical flow are concatenated along the channel dimension to form a joint representation $\mathbf{z} = [z_x, z_d, z_o]$. During training, we add noise to this joint latent at timestep ( $t \in [0, 1]$ ) to obtain $\mathbf{z}_t$. We extend the training objective to jointly predict all three modalities, defined as:

$$\mathcal{L}(\theta) = \mathbb{E}_{\mathbf{z}_1, \mathbf{z}_0 \sim \mathcal{N}(0, I), t \in [0,1]} \left[ \left| u^+ (\mathbf{z}_t, t; \theta) - v_t^+ \right|_2^2 \right], \qquad (2)$$

where $u^+ = [\hat{z}_x, \hat{z}_d, \hat{z}_o]$ denotes the predicted latents for RGB, depth, and optical flow, respectively, and $v_t^+ = [v_t^x, v_t^d, v_t^o]$ represents the target velocities. The loss is computed across the channels corresponding to the modalities. Unlike the complex sampling-stage guidance in VideoJAM [9], we adopt a simpler approach by simultaneously leveraging RGB and current perceptual conditions to guide future frame generation, as detailed in the next section.

### 3.3 Long-Horizon Generation Framework

Building on the unified modeling, we now describe how to effectively leverage historical frame contexts, including appearance and perceptual conditions, to achieve stable long video generation. Our design is informed by following key insights: (1) During training with teacher forcing [67], the model is conditioned on ground-truth frames as inputs. However, during inference, it must rely solely on its own previously generated frames. This discrepancy between training and inference can lead to the accumulation of errors over time. To mitigate this issue, we adopt a non-causal sampling head combined with delayed denoising [60], which allows the model to refine its predictions using additional context beyond the immediate past. (2) As shown in Fig. 3, depth prediction exhibits

greater temporal stability compared to RGB generation. When the model is trained to output only RGB, we observe rapid degradation in both color and shape over time. In contrast, depth outputs alone show significantly less drift. A key reason for this robustness is that depth is invariant to illumination, shadows, and surface textures, which introduce high variability in RGB signals and are difficult to model consistently over long sequences. This invariance reduces the visual complexity the model must reproduce, thereby slowing down error accumulation. Furthermore, jointly predicting RGB and depth substantially improves RGB quality, suggesting that the geometric constraints provided by depth help stabilize the appearance generation; (3) In contrast to the random per-frame noise scheduling used in Diffusion Forcing [10], we adopt a pre-defined, group-wise noise schedule that more closely aligns training with inference behavior. This structured scheduling strategy reduces training instability and improves overall efficiency by providing more consistent temporal dynamics across samples. Detailed results and comparisons are presented in Sec. 4.2.

Specifically, we partition the inputs to the Diffusion Transformer into two parts: *memory bank* and *prediction groups*. The memory bank consists of both short-term and long-term memory. For short-term memory, we select a small set of historical frames that are adjacent to the current frame. Since high-frequency details are typically reconstructed in the later denoising steps, we set the noise level of these frames to zero, allowing model to preserve fine-grained texture information. For long-term memory, these historical frames contribute less to the current frame's fine details. Although depth perception (as shown in Fig. 3) helps stabilize shape and size, RGB color drift can still occur. To bridge the training-inference gap, we apply a low-level noise $t_m$ to the RGB and optical flow channels of long-term memory, while keeping depth frames un-noised to provide stable global structural guidance. Without perceptual conditions, a high noise level (e.g., $t_m = 0.3$) is typically required for these channels, but this provides limited useful information. During training, $t_m$ is randomly sampled from 0.7 to 0.9 to enhance robustness, and during inference, it is set to 0.8. Note that a higher $t_m$ corresponds to a lower noise level, consistent with our preliminary.

For the prediction groups, we assign noise levels in a structured manner as follows: We divide the input frames into $G$ consecutive groups, where $G$ is the total number of groups. For each training sample, we sample an index $i$ uniformly from the interval $\left[1000 - \frac{1000}{G}, 1000\right)$. For each group $k$ ($k = 0, 1, \ldots, G-1$), the noise index is defined as:

$$\text{index}_k = i - k \cdot \frac{1000}{G}, \quad i \sim \mathcal{U}\left(1000 - \frac{1000}{G}, 1000\right) \tag{3}$$

The corresponding sigma and timestep for each group are obtained from the noise scheduler using $\text{index}_k$, the training timesteps is set to 1000. This scheduling ensures that the noise intervals are evenly distributed among the $G$ groups, and each group receives noise levels sampled from different segments of the scheduler curve. During inference, the groups are denoised sequentially in a streaming manner, with noise levels decreasing as each group is processed.

## 4 Experiments

### 4.1 Experimental Setup

**Datasets.** To comprehensively evaluate the effectiveness and generalizability of our proposed method, we conduct experiments using two publicly available foundation models: Wan2.1-1.3B [63] (flow-based), trained on an internal general-purpose video dataset at a resolution of $480 \times 832$; and CogVideoX-2B [72] (diffusion-based), trained on the in-the-wild DROID robotic manipulation dataset [42] at a resolution of $480 \times 720$. The general-purpose dataset contains 300K raw videos, segmented into 1 million short clips, while the DROID dataset consists of 200K successful robotic operation clips. This setup allows us to assess the feasibility of our approach on both flow-based and diffusion-based models, as well as its potential as a long-horizon world model in both general and specialized domains.

**Training.** All training is conducted on 32 NVIDIA A100 GPUs with a learning rate of 1e-4, using the AdamW optimizer [50], with a per-GPU batch size of 4. The Wan2.1-1.3B model is trained for 50K iterations while the CogVideoX-2B model is trained for 15K steps. To preserve text-to-video performance, we apply the same noise level to every frame in 10% of the training steps.

**Evaluation.** To assess video generation performance, we utilize VBench [36], a framework that evaluates models across disentangled dimensions, grouping metrics into two categories: consistency

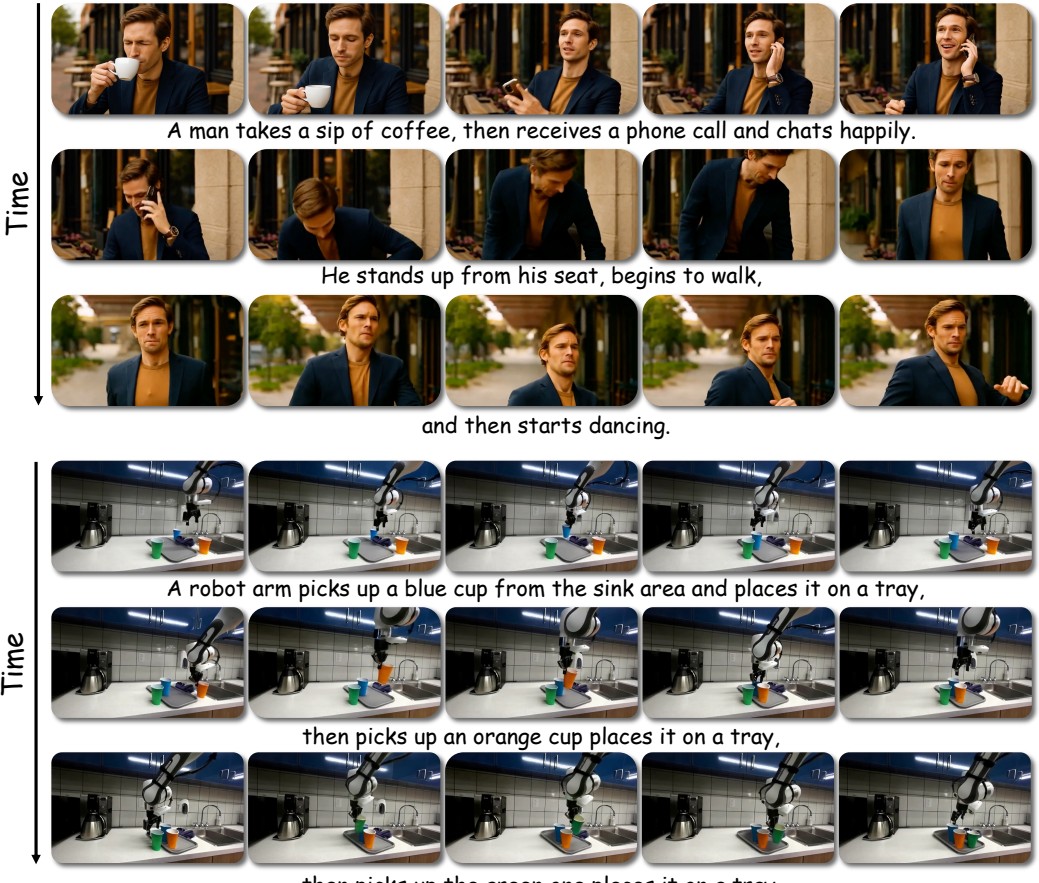

Figure 4: **Generated Long-Horizon Video Results of WorldWeaver.** WorldWeaver demonstrates strong generalization, as it can be applied to various base models. These results highlight not only the effectiveness of our method, but also its potential as a versatile and extensible world model across diverse domains.

and motion. Consistency metrics assess per-frame subject and background coherence, while motion metrics evaluate the amount and temporal coherence of generated motion. We also evaluate a start-end quality contrast metric to measure potential drifting. Specifically, we compute the absolute difference in image quality metric between the first 5 seconds and the last 5 seconds of each video. A larger value of $\Delta_{\text{drift}}^{\text{quality}}$ indicates a greater discrepancy between the beginning and end of the video.

## 4.2 Comparisons

**Visualization of WorldWeaver.** We present 30-second and 20-second long video generation results of our model on both general-purpose and robotic manipulation datasets in Fig. 4. Due to space limitations, additional visualizations and comparisons with existing methods are provided in the supplementary materials.

**Quantitative experiments.** We compare our model with existing open-source methods capable of generating long videos, including StreamingT2V [32], CausVid (Wan) [74], SkyReels-V2 1.3B [11], and Magi 4.5B [59]. Specifically, we use ChatGPT-4o to generate initial images and corresponding text prompts, each containing 4-6 actions, which are used to guide video generation. These prompts correspond to videos of 20-30 seconds in length. As shown in Tab. 1, WorldWeaver surpasses the Magi 4.5B on drift and background consistency metrics. These results are partly influenced by differences in training data and engineering optimizations. In the next section, we provide a more detailed comparison under the same settings. In addition, we conduct a user study to complement the automatic evaluation metrics, with further details available in the supplementary materials.

Table 1: **Comparison of with other open-source models.** Note that the motion degree metric is omitted, as nearly all 20-30 second videos are judged as true for this criterion, making it uninformative.

| Methods | Subject consistency | Background consistency | Image quality | Motion smoothness | $\Delta_{\text{drift}}^{\text{Quality}}$ |
|---|---|---|---|---|---|
| StreamingT2V | 79.34 | 84.32 | 0.47 | 0.73 | 0.23 |
| CausVid | 84.71 | 88.43 | 0.54 | 0.82 | 0.16 |
| SkyReels-V2 | 87.05 | 90.07 | 0.61 | 0.86 | 0.11 |
| Magi | **88.71** | 90.32 | **0.62** | **0.89** | 0.09 |
| Ours | 87.34 | **90.49** | 0.59 | 0.87 | **0.07** |

**Discussion of long context design.** Due to differences in the dataset, model architectures, training resources, and the use of post-processing techniques for long videos among existing open-source models, direct comparisons are not entirely fair. Therefore, We evaluate several relevant strategies for long-video generation using the same base model and dataset: *Repeating image-to-video (I2V)*, which applies zero noise to the first frame and identical random noise to all subsequent frames during training, then concatenates the model's final output back into the input sequence to extend video length; *CausVid* [74], which adapts the standard full-attention mechanism to a causal-only attention variant using the official implementation; *History Diffusion* [60], which feeds fully noised historical frames into the unconditional branch of classifier-free guidance to reinforce long-term memory; *Rolling Diffusion* [58], which applies a linearly decreasing, frame-wise noise schedule throughout the sequence; *Diffusion Forcing* [10], which applies random noise to each frame during training and delays the denoising of past frames during inference; and *Ours (RGB-only)* and *Ours (RGB+Perceptual)*, which use the memory bank and the prediction groups to model either RGB information alone or RGB and perceptual channels jointly. Since FramePack [76] requires pre-generating the first and last frames, it is not well suited for scenarios with substantial changes, such as robotic manipulation or dynamic scenes, and is thus not considered in our evaluation. It is worth noting that we also discuss the convergence time required for each algorithm, defined as the training duration after which further improvements in model performance are no longer observed.

Table 2: **Comparison with relevant long-horizon methods**. We compare with other relevant methods across the global metrics, drifting metric. The tests are conducted with CogVideoX-2B on the robotic manipulation dataset, with all experiments performed using 16 A100 GPUs.

| Methods | Subject consistency | Background consistency | Image quality | Motion smoothness | $\Delta_{\text{drift}}^{\text{Quality}}$ | Training steps |
|---|---|---|---|---|---|---|
| I2V | 90.23 | 91.36 | 0.51 | 0.64 | 0.24 | - |
| CausVid | 90.43 | 91.56 | 0.53 | 0.67 | 0.19 | - |
| History Diffusion | 89.78 | 91.92 | 0.57 | 0.68 | 0.09 | 70K |
| Rolling Diffusion | 88.74 | 91.09 | 0.58 | 0.70 | **0.05** | 20K |
| Diffusion Forcing | 88.67 | 91.23 | **0.61** | 0.72 | 0.06 | 80K |
| Ours *w/o* perceps | 89.57 | 91.51 | 0.55 | 0.67 | 0.15 | 30K |
| Ours | **90.92** | **92.39** | 0.60 | **0.75** | 0.07 | 30K |

As depicted in Tab. 2, our key findings are as follows: (1) Our approach achieves the best performance on most metrics, particularly in terms of consistency, while performing comparably to the best methods on other metrics. (2) Training with independent random noise schedules for each frame, which is adopted by Diffusion Forcing and History Diffusion, requires more training steps for convergence. In contrast, WorldWeaver reduced the training time with the improved training strategies with aligned training and inference noise schedules. (3) While delayed denoising of historical frames mitigates drift, the lack of clean historical frames as guidance compromises video consistency. (4) Compared to RGB-only modeling, unified modeling with perceptual conditions enhances per-frame prediction accuracy, allowing the historical frames of the memory bank to maintain a low noise level, thereby providing effective guidance for subsequent predictions and improving overall consistency.

### 4.3 Ablation studies

**Impact of perceptual conditions.** We assess the contributions of various perceptual conditions to improving different aspects of video generation. In addition to the video depth and optical flow

adopted in our final framework, we also explore joint modeling with video segmentation, which is derived using SAM2 [55] by retaining up to 10 masks with the highest confidence scores for each video. To eliminate the influence of drift, we conduct these experiments on short 5-second videos.

Table 3: **Effects of various perceptions.** We evaluate the impact of combining RGB with individual perceptual conditions, as well as jointly using multiple perceptual conditions together.

| Perceptions | Subject consistency | Background consistency | Dynamic degree | Motion smoothness |
|---|---|---|---|---|
| - | 92.72 | 94.76 | 0.62 | 0.74 |
| *w/* depth | 94.87 | 95.14 | 0.70 | 0.82 |
| *w/* seg | 94.01 | 94.83 | 0.65 | 0.70 |
| *w/* flow | 92.48 | 94.22 | **0.74** | 0.81 |
| *w/* depth & seg | **94.93** | 95.17 | 0.68 | 0.77 |
| *w/* depth & flow | 94.76 | **95.23** | 0.73 | **0.85** |

As shown in Tab. 3, we find that video depth improves both consistency and motion, while optical flow mainly enhances motion. Segmentation contributes more to consistency but slightly reduces motion quality. Since both depth and segmentation capture similar shape and size information, combining them does not further improve consistency. In contrast, jointly modeling depth and optical flow leads to better motion performance. Compared to VideoJAM [9], which relies primarily on optical flow with complex sampling guidance, our approach demonstrates that conditioning on video depth yields greater improvements in temporal coherence and structural fidelity, with optical flow providing complementary benefits.

**Noise level of memory bank.** We further analyze the impact of the noise levels applied to the long-term memory. As demonstrated in Tab. 4, when perception conditions are not applied, reducing the noise level in the memory bank improves the overall temporal continuity of the video, mitigating abrupt changes in certain frames. However, due to the gap between the generated video during inference and the ground-truth video during training, accumulated errors exacerbate video drift. In contrast, incorporating perception conditions enhances model robustness, allowing clearer past frames to be input into the model as guidance with much lower quality drift. We randomly vary the noise level $t_m$ for RGB and flow latents between 0.7 and 0.9 during training to enhance robustness, and set it to a fixed value of 0.8 during inference. Note that a higher $t_m$ corresponds to a lower noise level, consistent with our preliminary.

Table 4: **Ablation study on noise level.** Adding noise to the memory bank reduces drift but reduces consistency. With perceptual conditions, the model is more robust to drift and less sensitive to noise level. The tests are conducted with CogVideoX-2B on the robotic manipulation dataset.

| $t_m$ | Subject consistency | | Background consistency | | Image quality | | Motion smoothness | | $\Delta_{drift}^{Quality}$ | |
|---|---|---|---|---|---|---|---|---|---|---|
| | *w/o* | *w/* | *w/o* | *w/* | *w/o* | *w/* | *w/o* | *w/* | *w/o* | *w/* |
| 0.9 | 89.64 | **91.78** | 92.17 | **92.56** | 0.52 | 0.60 | 0.67 | 0.73 | 0.19 | 0.07 |
| 0.7 | 89.23 | 90.17 | 91.45 | 92.37 | 0.56 | **0.62** | 0.69 | **0.76** | 0.15 | 0.06 |
| 0.3 | 88.49 | 88.98 | 90.62 | 91.34 | 0.60 | 0.61 | 0.70 | 0.75 | 0.10 | **0.04** |

**Ablation on number of groups and length of memory bank.** To further analyze the impact of memory bank length and the number of prediction groups, we conduct an ablation study on these hyperparameters. Let $F$ denote the total number of input latent frames, $G$ the number of prediction groups, $N$ the number of frames per group, and $M$ the memory bank length. These parameters are related by $N \cdot G + M = F$. We fix $F = 21$ for all settings, in order to fix the computational cost. Within the memory bank, one frame is reserved for short-term memory (fully denoised frame), while the remaining $M - 1$ frames are dedicated to long-term memory. As shown in Tab. 5, considering the overall performance across all metrics as well as the number of frames generated under the same computation, we set $M = 5$ and $N = G = 4$ in our final training configuration. This experiment is conducted on the robotic manipulation dataset using 16 A100 GPUs, consistent with the ablation studies in the main paper, including the discussion on methods for long videos and the analysis of

Table 5: **Ablation study.** Notations such as `m1_g4_n5` indicate a memory bank length $M = 1$, prediction groups $G = 4$, and frames per group $N = 5$, and so forth, while `Frameoutput` refers to the number of video frames produced by a single complete denoising pass.

| Setting | Subject consistency | Background consistency | Image quality | Motion smoothness | $\Delta_{drift}^{Quality}$ | frame output |
|---|---|---|---|---|---|---|
| `m1_g4_n5` | 88.85 | 91.30 | 0.61 | **0.75** | **0.05** | 20 |
| `m5_g4_n4` | 90.92 | 92.39 | 0.60 | **0.75** | 0.07 | 16 |
| `m13_g4_n2` | **91.04** | **92.51** | 0.58 | 0.74 | 0.09 | 8 |
| `m5_g8_n2` | 90.64 | 91.87 | **0.62** | 0.73 | 0.06 | 16 |
| `m1_g2_n10` | 90.55 | 91.23 | 0.56 | 0.68 | 0.12 | 20 |

noise levels in the memory bank. For these analyses, we extract the first frame from videos where DROID operations fail, using these frames as input images. This subset of data is entirely unseen during training. Based on these first frames, we employ ChatGPT o3 to generate 30 prompts for 15–20 second robotic operations, each containing 3–5 action commands (e.g., move, grasp, switch, push/pull). For the ablation study on the contribution of perceptual conditions, we use 60 short 5-second videos to isolate the effects from long video frameworks.

**Drift resistance experiment details.** To investigate the drift resistance of perceptual conditions and color information, we conduct the following experiment after model training. For the depth-only and RGB-only variants, we retain only the weights corresponding to the relevant input and output channels, while loading all other parameters from the fully trained model. We then select a single image or depth map and replicate it across 81 frames (to match the model's input sequence length), appending "static" to the training caption. This process generates 10,000 such samples, which we fine-tune for 1,000 steps each, enabling the model to perform a simple task: consistently outputting static images in a streaming manner. Notably, since optical flow represents motion between frames and lacks temporal continuity as a signal, its isolated output lacks meaningful consistency, so we exclude it from this experiment. We assess the drift resistance by measuring the normalized Mean Squared Error (MSE) between the subsequently generated frames and the first generated frame. As shown in Fig. 5, depth output alone demonstrates superior drift resistance. Jointly outputting both RGB and depth mitigates the drift phenomenon in RGB.

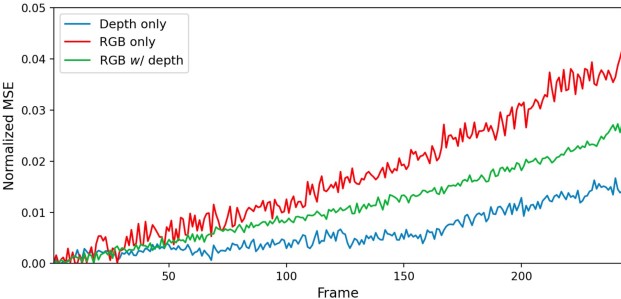

Figure 5: **Normalized mse for drift resistance.**

## 5 Conclusion

We present WorldWeaver, a framework for long video generation that jointly model RGB information and perceptual conditions under a unified long-context modeling pipeline. We find that jointly predicting RGB and perceptual signals leads to improvements in both consistency and motion quality. Leveraging perceptual conditions, especially depth, which is more resistant to drift than color, enables us to better preserve historical context and enhance temporal consistency. Furthermore, by adopting non-causal prediction groups and a group-wise noise strategy that ensures alignment between training and inference, we are able to alleviate drift and reduce the overall training cost. Extensive experiments on robotic-manipulation and in-the-wild datasets show that WorldWeaver improves long-horizon stability and visual fidelity. It achieves superior consistency and drift mitigation, confirming its effectiveness.

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

## Contents

## 6 Additional Visual Results

### 6.1 More visualizations

As shown in Fig. 6, we present additional videos generated by WorldWeaver. Additionally, we recommend accessing the *viewer.html* file for a visual comparison between our approach and current state-of-the-art methods

## 7 User Study

To conduct a comprehensive comparison, we evaluate our model (based on Wan2.1 1.3B [63]) against recently released state-of-the-art long-video models of similar scale, including SkyReels-V2 1.3B [11] and MAGI 4.5B [59]. Specifically, we use 48 prompts, each containing 4–6 actions, to generate videos lasting 20–30 seconds. All prompts are provided in the `prompts.txt` file. We engage 15 annotators to complete a questionnaire, with the UI screenshot presented in Fig. 7. The questionnaire comprises five questions: (1) Which video has the highest overall image quality? (2) Which video exhibits better consistency (considering both subject and background)? (3) Which video shows the smallest difference between the first 5 seconds and the last 5 seconds (in terms of quality and

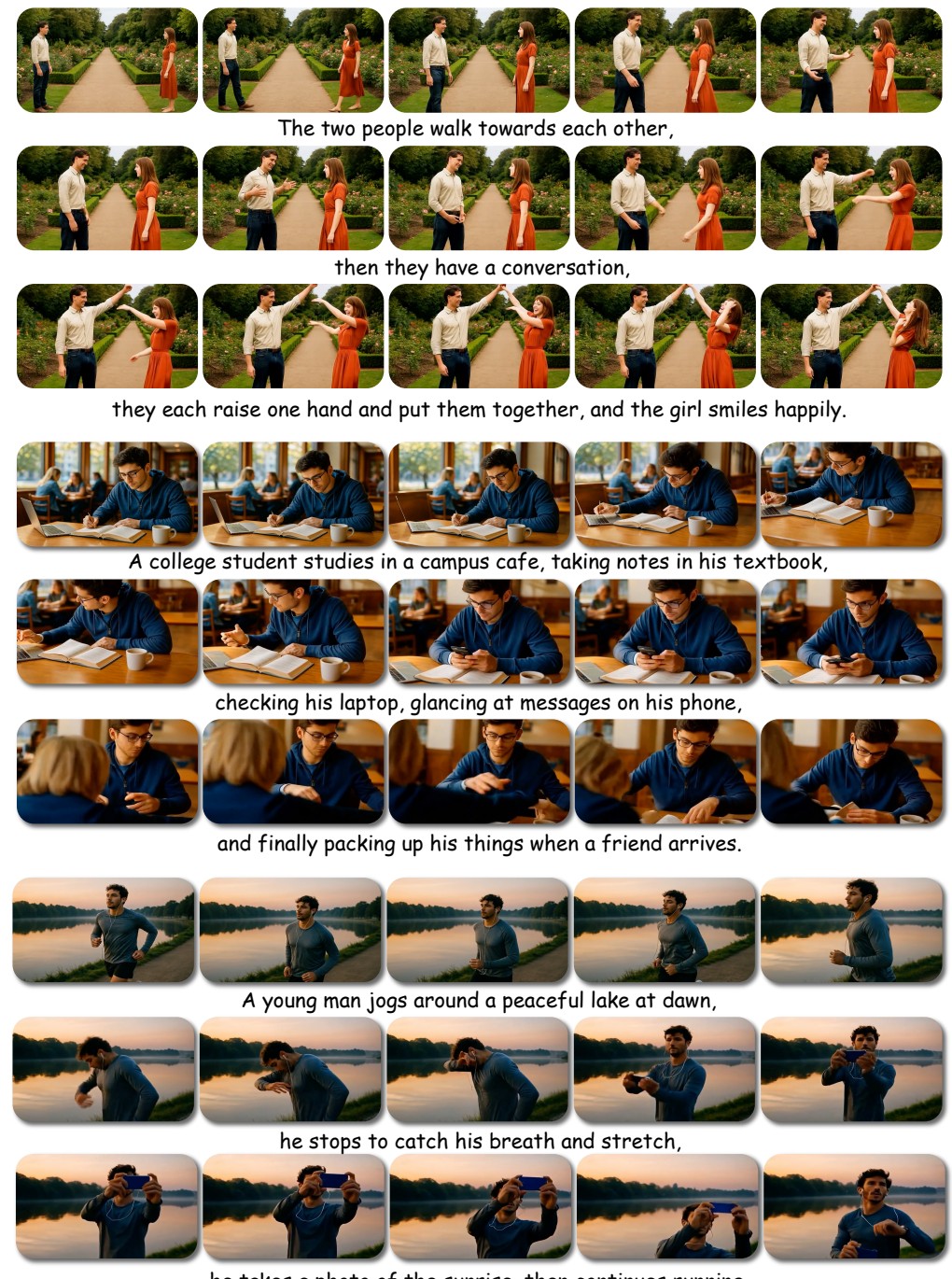

The two people walk towards each other,

then they have a conversation,

they each raise one hand and put them together, and the girl smiles happily.

A college student studies in a campus cafe, taking notes in his textbook,

checking his laptop, glancing at messages on his phone,

and finally packing up his things when a friend arrives.

A young man jogs around a peaceful lake at dawn,

he stops to catch his breath and stretch,

he takes a photo of the sunrise, then continues running.

Figure 6: **More visualizations.**

consistency)? (4) Which video has smoother motion? (5) Which video aligns best with the overall action described in the prompt.

Results are shown in Fig. 8. Our method demonstrates performance comparable to MAGI in motion smoothness and consistency metrics, while achieving superior results in quality drift. However, a slight gap in image quality remains, with overall performance generally surpassing SkyReels-V2. These findings are generally consistent with our measurements on the VBench benchmark.

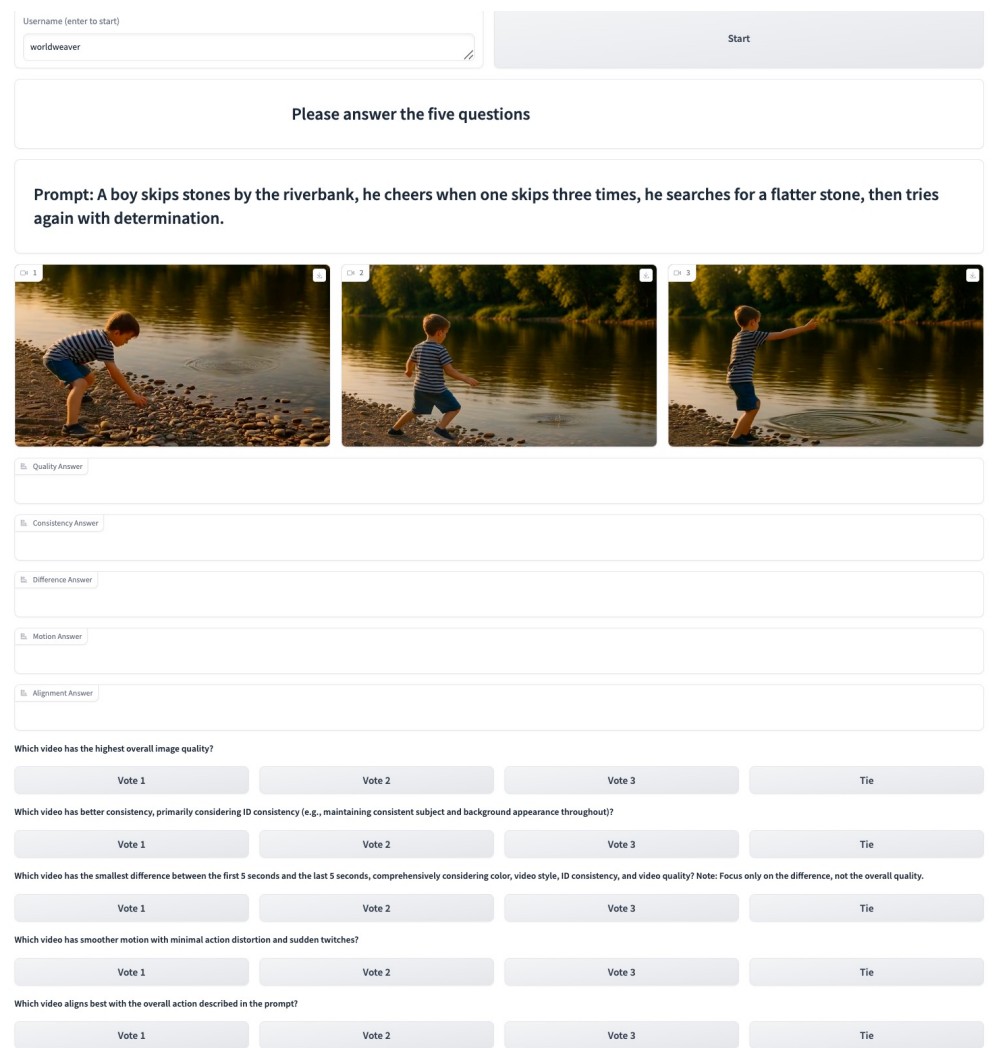

Figure 7: **Visualization of the user study interface.**

## 7.1 Limitations and future works

**Limitations and future works.** Despite its strengths, our work has several limitations. First, learning stable physical dynamics from the complexity of the real world remains a long way off, and our model is far from perfect. Operations involving very small objects can still exhibit sudden disappearances, since even depth-based cues cannot fully capture these small objects. Second, although fine-tuning existing models allows us to generate 10∼20s videos, the success rate of generation still decreases as the video length increases, and error accumulation persists over longer horizons, which we leave as an important direction for future work. Finally, while video depth is shown to be the most effective perceptual condition in our experiments, investigating additional or complementary cues to further exploit the rich information in real-world data also represents a promising avenue for further study. **Broader impacts.** As video generation technology continues to advance, the development of robust authentication and forgery detection methods becomes increasingly critical. The ability to create highly realistic videos, such as those generated by our model, underscores the need for parallel advancements in counterfeit identification to mitigate potential misuse, ensuring the integrity of digital content in an era of rapid technological evolution.

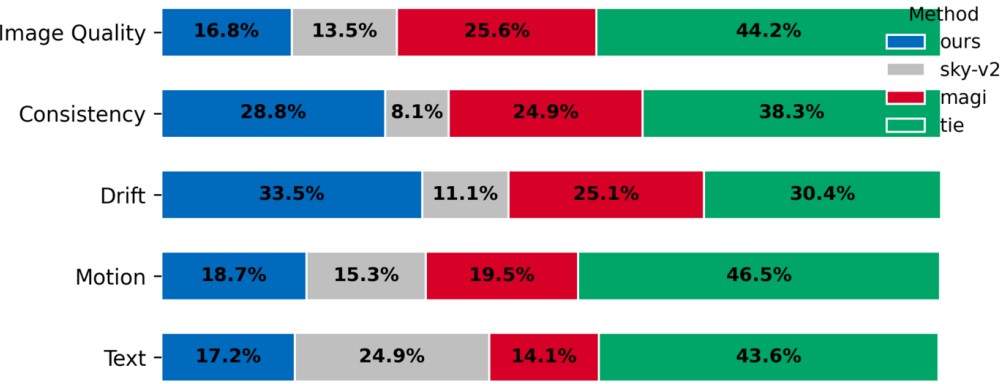

Figure 8: **Results of user study.**

