# OpenReview forum: "WorldWeaver: Generating Long-Horizon Video Worlds via Rich Perception"
_NeurIPS.cc/2025/Conference — NeurIPS 2025 poster_

### Official Review · Reviewer_6nHu · 2025-07-01

**Clarity:** 2
**Significance:** 3
**Originality:** 3
**Rating:** 4
**Confidence:** 3

**Summary:**

This work tackles the task of long video generation, with three contributions: 1) a joint modeling framework that predicts RGB, depth and optical flow; a perceptual memory bank with depth to record historical frames. 3) A new segmented noise scheduling strategy that reduces the training time. Extensive experiments show improved subject / background consistency and motion smoothness on general-purpose and robotic benchmarks.

**Questions:**

- The depth and optical flow signals for in-the-wild videos are predicted using off-the-shelf models, which may introduce unaligned signal pairs, inconsistent depth signals and hence accumulated errors. Is there a mechanism to correct or mitigate these errors?
- What's the benefit of training with the internal general-purpose dataset? It would be good to see results that train with publically available general-purpose datasets for more thorough evaluation.
- Evaluation-wise, the proposed method is only tested on generated images and texts. How is the performance on in-the-wild test inputs?

**Ethical Concerns:**

["NO or VERY MINOR ethics concerns only"]

**Final Justification:**

The authors' response addresses most of my concerns. Thus, I tend to keep my original score and lean towards acceptance.

**Limitations:**

Yes

**Paper Formatting Concerns:**

None.

**Quality:**

3

**Strengths And Weaknesses:**

Strengths:
+ Introducing auxiliary perceptual signals depth and optical flow in the long video generation framework is new and proves to maintain consistent motion across long context.
+ Ablative experiments prove the reduced training epochs and the effectiveness of the perceptual memory bank and the group-wise
 noise scheduling.

Weakness:
- Paper organization looks a bit off. Section 3.3 is better to be separated into two subsections for each contribution.
- The depth and optical flow signals for in-the-wild videos are predicted using off-the-shelf models, which may introduce unaligned signal pairs, inconsistent depth signals and hence accumulated errors. Is there a mechanism to correct or mitigate these errors?
- What's the benefit of training with the internal general-purpose dataset? It would be good to see results that train with publically available general-purpose datasets for more thorough evaluation.
- Evaluation-wise, the proposed method is only tested on generated images and texts. How is the performance on in-the-wild test inputs?
- There are mainly static illustrations and only three side-by-side qualitative video comparisons in the paper. It would be better to show more dynamic cases to evaluate the motion quality and temporal consistency.
- The presented results still contain temporal inconsistency at semantic transitions (supp html case 1, the woman's expression changes from none to laughing is unnatural).

---

> ### Author Rebuttal · Authors · 2025-07-30
>
> ### **Author Response to Reviewer 6nHu**
> We sincerely appreciate your recognition and thoughtful feedback, which are instrumental in refining and strengthening our work. We hope the following response adequately addresses your question.
>
>  **Q1: Clarifying the paper organization.**
>
>  **A1:** Thank you for pointing this out. In the revision we will split Section 3.3 into two distinct subsections:  (1) Perceptual Memory Bank (design, motivation and ablations), and (2) Group‑wise Noise Scheduling (training/inference alignment, efficiency). This re‑structure cleanly separates the two contributions and improves readability without altering technical content.
>
>  **Q2: Depth/Flow mis‑alignment mitigation.**
>
> **A2:** Thank you for the question. For depth we apply an affine‑invariant normalization to convert raw depth maps into relative depths in \([-1,1]\), following Marigold[1]:
>
> $$
> \tilde{d} = \left(\frac{d - d_{2}}{d_{98} - d_{2}} - 0.5\right)\times 2
> $$
>
> where $ d_{2} $ and  $ d_{98} $ are the 2nd and 98th percentiles of each frame’s depth distribution. This removes extreme outliers and global scale/shift variations so the model focuses on relative spatial relationships.
>
> For optical flow we divide each displacement vector by the frame diagonal, $\sqrt{H^2 + W^2}$. Since all training clips use the same resolution, this aligns motion magnitudes across the dataset. The normalized flow is then encoded as an RGB video (hue = direction, opacity = magnitude).
>
>  **Q3: Benefits of internal datasets.**
>
> **A3:** Our task requires full-model fine-tuning rather than freezing the backbone or applying lightweight adaptation methods such as LoRA. However, most large-scale publicly available datasets contain various quality issues—such as embedded subtitles, watermarks, or overly chaotic motion patterns—which significantly affect the training process. Notably, existing models like the pretrained Wan, Magi, and SkyReels-V2 have all undergone a final-stage supervised fine-tuning (SFT) on carefully curated high-quality data. If we were to further fine-tune such models on noisy public data, it would significantly degrade video quality and obscure the true contribution of our modeling framework. In contrast, our internal general-purpose dataset provides relatively higher data quality—with better control over subtitles, watermark removal.
>
> We fully agree that more thorough evaluation across diverse datasets is important. As presented in Table 2 of the main paper, we have already conducted a comparison across multiple methods using the same public robotic manipulation dataset. However, due to the inherent difficulty of reliably cleaning and filtering large-scale public video datasets—e.g., removing subtitles, watermarks, and abrupt transitions—we adopted a controlled setting using our internal general-purpose dataset. Specifically, we re-trained multiple baseline methods on the same internal dataset to ensure fair and consistent evaluation across architectures. This setup allows us to isolate the effect of model design without the confounding influence of data quality variation. We will also include additional results on public general-purpose datasets in the revised version to further demonstrate the robustness and generalization ability of our method.
>
> | Methods         | Subject consistency | Background consistency | Image quality | Motion smoothness | Δ Quality drift |
> |-----------------|---------------------|-------------------------|---------------|-------------------|-----------------|
> | CausVid  |  87.06             | 89.68                  | 0.55          |    0.83          |  0.11         |
> | DF  |  86.72                | 89.42                  | 0.56         |    0.86          |  0.07       |
> | Ours *w/o* perceps   |  86.81                | 89.54                  | 0.55         |    0.83          |  0.12         |
> | Ours   |  87.34                 | 90.49                  | 0.59          |    0.87          |  0.07         |
>
>  **Q4: Test on  in-the-wild inputs.**
>
> **A4:** We thank the reviewer for their valuable feedback regarding the evaluation on in-the-wild inputs. In our experimental design, we utilized text prompts and initial images generated by a third-party tool (ChatGPT). This approach was intentionally chosen to establish a fair and standardized benchmark for all compared methods, thereby eliminating potential biases that could arise from manually curated prompts.
>
> Regarding the model's performance on in-the-wild data, our method exhibits strong robustness. This is because the model itself was trained on large-scale, in-the-wild datasets. Furthermore, our qualitative results on the robotic manipulation dataset are  generated from real-world scenarios.
>
> We will incorporate extensive results on in-the-wild videos in the final revision to demonstrate our framework's robustness, as we are limited from showing new results during the rebuttal period.
>
>  **Q5: Dynamic cases.**
>
> **A5:** We agree with the reviewer on the importance of showcasing more dynamic cases, particularly those with complex motions and semantic transitions, and will incorporate them into the final version. We candidly acknowledge that while our joint training approach improves upon RGB-only models, generating long videos with large motion (like long dance sequences) remains a common challenge for the field. This is largely due to error accumulation in streaming frameworks, where a single degraded frame can impact the entire subsequent sequence.
>
> Our method helps to mitigate this issue. By jointly modeling RGB with stable perceptual cues like depth and flow, it generates a more reliable initial context, thus providing a more robust basis for subsequent frames.
>
> [1] Ke, Bingxin, et al. "Repurposing diffusion-based image generators for monocular depth estimation." Proceedings of the IEEE/CVF conference on computer vision and pattern recognition. 2024.

---

> > ### Comment · Reviewer_6nHu · 2025-08-05
> >
> > I would like to thank the authors for the detailed response, which addressed most of my concerns. Thus, I tend to keep my original score and lean towards acceptance. Please include the discussed qualitative results on in-the-wild, dynamic cases in the final version.

---

> > > ### Author Response · Authors · 2025-08-06
> > >
> > > We appreciate you keeping your score and leaning toward acceptance! We will include the discussed qualitative results on in-the-wild, dynamic cases in the final version.

---

### Official Review · Reviewer_bGie · 2025-07-03

**Clarity:** 3
**Significance:** 2
**Originality:** 2
**Rating:** 4
**Confidence:** 4

**Summary:**

The paper WorldWeaver targets the drifting (error accumulation) and gradual structure‑deformation problems of autoregressive generation models. The authors propose to use multi‑modal prediction, including RGB, monocular depth (VideoDepth‑Anything) and optical flow (SEA‑RAFT) (all encoded by a 3D VAE and concatenated). The DiT/Flow‑Matching backbone is fine‑tuned to regress jointly for all these channels. It also uses a perceptual memory bank, which is a sliding window of past frames is split into short‑term (noise‑free) and long‑term memories; the RGB+flow part of long‑term memory is mildly noised while depth remains clean, under the hypothesis that geometry is more drift‑resistant. The authors also propose to do segmented group‑wise noise schedule aligned between training and inference; groups are denoised sequentially with delayed denoising. Significant improvement can be observed from the generated videos comparing with Magi-1 and SkyReels-v2, two SoTA autoregressive video generation models.

**Questions:**

N/A

**Ethical Concerns:**

["NO or VERY MINOR ethics concerns only"]

**Final Justification:**

I do not have much problem with this paper initially and would keep my score.

**Quality:**

3

**Strengths And Weaknesses:**

Strength:

-- The motivation is clear: RGB‑only prediction has bias and this is a good motivation for multi‑perception conditioning, which is supported by toy “static scene drift” experiment (Fig. 3).

-- The authors provide comprehensive different types of multi-modal inputs and outputs, which are thoroughly considered.

-- The multi-modal memory‑bank design grounded in empirical observation that depth drifts slower; elegant to keep depth clean and RGB slightly noised.

-- Comprehensive ablations are provided on perception types, memory‑bank noise, and training noise schedule.


Weakness:

-- The idea of including multi-modality inputs and outputs are relatively straightforward and has been discussed (e.g. VideoJAM at ICML'25), and its "memory bank" is more of a diffusion forcing variant when dealing with more than RGB channels. The novelty is arguably a bit incremental but I think the authors executed everything well.

---

> ### Author Rebuttal · Authors · 2025-07-30
>
> ### **Author Response to Reviewer bGie**
> We sincerely appreciate your valuable feedback and your positive assessment of our work. Below, we address your comments in detail.
>
>  **Q1: Clarifying the contributions beyond existing methods.**
>
>  **A1:** While our work is partially inspired by VideoJAM, our contributions are distinct in two key aspects.
> - First, we systematically analyze different perceptual conditions and find that video depth offers more comprehensive improvements —enhancing both motion quality and temporal consistency—compared to optical flow. Its robustness to visual drift is a key insight that motivates our design.
> - Second, we address a fundamental tension in prior methods like Diffusion Forcing: low noise in historical frames leads to drift, while high noise limits useful context. We resolve this by leveraging depth’s stability and proposing a memory-augmented framework that retains clearer historical cues. This allows effective use of long-term context without sacrificing consistency.

---

> > ### Comment · Reviewer_bGie · 2025-08-05
> >
> > I do not have many concerns initially and would keep my positive score.

---

> > > ### Author Response · Authors · 2025-08-06
> > >
> > > Thank you for the positive assessment and for keeping your score. We appreciate the support and will continue to refine the paper!

---

### Official Review · Reviewer_vMBw · 2025-07-03

**Clarity:** 3
**Significance:** 3
**Originality:** 2
**Rating:** 4
**Confidence:** 4

**Summary:**

This paper proposes WorldWeaver, a framework for long-horizon video generation aimed at addressing the challenges of structural and temporal consistency in generative video modeling. The key contributions include predicting perceptual conditions (depth and optical flow) alongside RGB frames using the same model, implementing a memory bank to preserve context through depth stability, and introducing segmented noise scheduling to mitigate temporal drift. The authors validate their method using both two video models on one general-purpose dataset and one robotic-manipulation dataset, demonstrating improvements over baseline approaches in consistency and drift metrics.

**Questions:**

1. How robust is the video depth processing pipeline?

2. How will the depth prediction quality affect generative performance?

**Ethical Concerns:**

["NO or VERY MINOR ethics concerns only"]

**Final Justification:**

Most of my concerns have been addressed. I believe the paper is above the acceptance bar despite the concern of novelty. However, the messages conveyed by the paper, e.g., depth > optical flow for long video generation, are useful for the community. Thus, I would like to keep my original rating.

**Limitations:**

Yes

**Quality:**

3

**Strengths And Weaknesses:**

### Strengths

- This work leverages carefully curated video depth predictions (using Video Depth Anything and Depth Anything V2) alongside optical flow predictions to effectively mitigate temporal drift and structural degradation, issues commonly observed in video generative models.

- It introduces a memory bank strategy that maintains noise-free depth information and slightly noised RGB and optical flow information, enhancing model robustness.

- To further improve upon existing methods like diffusion forcing and history guidance, the paper proposes a segmented noise scheduling strategy, specifically alleviating exposure bias issues in long-sequence video generation.

- Overall, the paper's presentation and clarity are commendable, despite some minor issues.


### Weakness
- Limited novelty.
    - First of all, I understand this field is crowded and it is hard to study something unexceptedly new or interesting.
    - But this work mainly primarily builds upon and extends ideas from existing works, particularly VideoJAM, by adding depth as additional input&output. Furthermore, the techniques proposed to handle exposure bias are incremental improvements of known methods like those from diffusion forcing / history-guidance, noisy-context learning works.

- On comparison and ablation studies.
    - Most experiments, apart from those in Table 1, are conducted exclusively on the robotic manipulation dataset, DROID (I didn't find what dataset Table 3 is using). In this dataset, camera motion is absent and object movements are relatively fixed. The observations and conclusions drawn from robotic datasets hardly transfer to general-purpose datasets. The authors should include more experiments on general-purpose datasets to strengthen their claims.
    - The definition of "converged models" used for comparisons in ablation studies (mentioned in line 259) is unclear. Given that lower loss in flow matching or diffusion training does not necessarily imply better generative performance in many cases, it is essential to allocate equal computational resources to each method to ensure fair comparisons.
    - In Table 1, it would be highly beneficial to include a baseline that employs the same base model as the proposed method, potentially using a repeated image-to-video extension strategy if direct long-video synthesis is not feasible. This comparison would more clearly indicate how much performance improvement stems directly from the proposed components.

Minors:
- It would be helpful to provide a more comprehensive preliminary section for a broader audience, such as delayed denoising.
- Given the critique of VideoJAM’s "complex sampling-stage guidance," it is necessary for the authors to clarify why the guidance approach in VideoJAM is considered complex and how the proposed method simplifies or improves upon it.
- There're many typoes in intro, e.g., "strategyassinignig", "inferenceto", perceptual-conditioned "next prediction", and more. Please fix them.


I'm happy to raise my score.

---

> ### Author Rebuttal · Authors · 2025-07-30
>
> ### **Author Response to Reviewer vMBw**
> We sincerely appreciate your kind words about our work. We believe the response below clarifies your concern.
>
>  **Q1: Limited novelty.**
>
>  **A1:**  Thank you for your feedback. While our work is partially inspired by VideoJAM, our contributions are distinct in two key aspects.
> - First, we systematically analyze different perceptual conditions and find that video depth offers more comprehensive improvements —enhancing both motion quality and temporal consistency—compared to optical flow. Its robustness to visual drift is a key insight that motivates our design.
> - Second, we address a dilemma in prior methods like Diffusion Forcing: low noise in historical frames leads to drift, while high noise limits useful context. We resolve this by leveraging depth’s stability and proposing a memory-augmented framework that retains clearer historical cues. This allows effective use of long-term context without sacrificing consistency.
>
>  **Q2: Ablation study on general-purpose datasets.**
>
>  **A2:** We completely agree with your point that evaluation on general-purpose datasets is essential for validating broader applicability. Due to computational cost constraints, we selected a subset of baselines—CausalVid, Diffusion Forcing, as well as our own framework without perceptual conditioning—and retrained them on the same general-purpose dataset using the Wan-1.3B backbone for direct comparison. These results are consistent with the findings reported in the main paper.
>
> | Methods         | Subject consistency | Background consistency | Image quality | Motion smoothness | Δ Quality drift |
> |-----------------|---------------------|-------------------------|---------------|-------------------|-----------------|
> | CausVid  |  87.06             | 89.68                  | 0.55          |    0.83          |  0.11         |
> | DF  |  86.72                | 89.42                  | 0.56         |    0.86          |  0.07       |
> | Ours *w/o* perceps   |  86.81                | 89.54                  | 0.55         |    0.82          |  0.12         |
> | Ours   |  87.34                 | 90.49                  | 0.59          |    0.87          |  0.07         |
>
>  **Q3: Definition of Converged Models.**
>
>  **A3:**  Thank you very much for pointing this out. We used equal computational resources and training configurations for all methods to ensure fairness, except for the total number of training steps, which was intentionally varied to investigate the convergence behavior of different algorithms. In practice, we do not rely solely on loss values to determine convergence. Instead, we monitor both quantitative evaluation metrics and visual results at inference time. Once both have stabilized and no further improvement is observed, we consider the model to be converged and record the corresponding number of training steps. We also continue training beyond the observed convergence point to ensure that no further improvements can be achieved. We will clarify this definition explicitly in the revised version.
>
>  **Q4: Lack of a fair baseline using same base model.**
>
>  **A4:** Thank you for the insightful suggestion. Our method is based on a 1.3B text-to-video (T2V) model, while Wan does not provide a corresponding 1.3B image-to-video model. As an alternative, we use the Wan-14B image-to-video model for evaluation on the same benchmark. It is worth noting that these pretrained model baselines have undergone high-quality supervised fine-tuning (SFT), whereas our training data is not as high-quality as the curated SFT data.
>
> As shown in the table below, we observe that increasing model size does lead to improvements in overall video quality and motion scores. Nevertheless, these larger models still exhibit more pronounced temporal drift. This is largely due to the repeated image-to-video generation strategy, which lacks access to prior context and suffers from a severe training-inference gap.
>
>
> | Methods         | Subject consistency | Background consistency | Image quality | Motion smoothness | Δ Quality drift |
> |-----------------|---------------------|-------------------------|---------------|-------------------|-----------------|
> | WanI2V 14B    | 89.86               | 91.02                 | 0.63          |       0.90        | **0.15**          |
> | Ours 1.3B    |  87.34                 | 90.49                  | 0.59          |    0.87          |  0.07         |
>
> Despite using a smaller 1.3B model, our method significantly mitigates temporal drift and improves long-horizon video quality through its architectural and training design. Moreover, as presented in Table 2 of the main paper, we compare repeated image-to-video generation with other autoregressive and streaming baselines on the same dataset, and consistently find that autoregressive generation is more suitable for long-horizon video synthesis.
>
>
>  **Q5: Clarification.**
>
>   **A5.1 Preliminary:** Thank you for the suggestion. We will expand the preliminary section in the revised version to better support a broader audience
>
>  **A5.2 Complexity of VideoJAM's sampling guidance:** VideoJAM’s sampling guidance is considered complex for two main reasons. First, it requires additional forward passes during inference, which increases computational overhead. Second, it introduces multiple guidance scales as additional inference-time parameters. These parameters can affect generation quality and may require manual adjustment depending on the scene or prompt.
>
> **A5.3 Typos:** We will carefully proofread the paper and correct all typographical and formatting errors in the revised version.
>
> **A5.4 Depth estimation robustness:** Our depth pipeline is based on state-of-the-art models (Video Depth Anything), which are robust across a wide range of scenes, making the overall depth processing pipeline reliable in practice. Crucially, our method does not rely on precise metric depth. We apply an affine‑invariant normalization to convert raw depth maps into relative depths in \([-1,1]\), following Marigold[1]:
>
> $$
> \tilde{d} = \left(\frac{d - d_{2}}{d_{98} - d_{2}} - 0.5\right)\times 2
> $$
>
> where $ d_{2} $ and  $ d_{98} $ are the 2nd and 98th percentiles of each frame’s depth distribution. This removes extreme outliers and global scale/shift variations so the model focuses on relative spatial relationships. Depth serves as a relative structural prior, providing stable geometric cues that help maintain temporal consistency and stabilize long-horizon generation—which current models are sufficient to support.
>
> [1] Ke, Bingxin, et al. "Repurposing diffusion-based image generators for monocular depth estimation." Proceedings of the IEEE/CVF conference on computer vision and pattern recognition. 2024.

---

> ### Comment · Reviewer_vMBw · 2025-08-05
>
> I thank the authors for their detailed response. Most of my concerns have been addressed. I believe the paper is above the acceptance bar despite the concern of novelty. However, the messages conveyed by the paper, e.g., depth > optical flow for long video generation, are useful for the community. Thus, I would like to keep my original rating. Please include these results during the rebuttal phase into the paper.

---

> > ### Author Response · Authors · 2025-08-06
> > **Comment by Authors**
> >
> > Thank you for the thoughtful follow-up and positive assessment! We’re glad most concerns have been addressed and appreciate your view that the paper is above the bar.

---

### Note · Authors · 2025-08-11

We sincerely thank all reviewers for their constructive feedback and the unanimously positive initial scores. We are pleased that our rebuttal successfully addressed the reviewers’ concerns, and that, in the final discussion stage, all reviewers continued to lean toward acceptance. We also thank the AC for their efforts in coordinating the review process and facilitating constructive discussions.

WorldWeaver offers a unified framework that jointly models RGB and perceptual conditions with a memory-augmented design and segmented noise scheduling, significantly improving temporal consistency and motion quality. Our comprehensive ablations and comparisons on both general-purpose and robotic benchmarks, along with additional evaluations promised in the final version, further demonstrate the robustness and generality of our approach.

We appreciate the thoughtful suggestions on clarity, dataset diversity, and qualitative results, and will incorporate these improvements in the final version. We believe WorldWeaver will provide a practical and extensible foundation for future advances in long-horizon video generation.

---

### Decision · Program_Chairs · 2025-09-17

**Decision:**

Accept (poster)

**Comment:**

This paper addresses long-horizon video generation by jointly predicting auxiliary signals (depth and optical flow) alongside RGB, and introduces a perception-conditioned framework with a memory bank and group-wise noise scheduling to preserve context and reduce drift.

The paper initially received three borderline accepts. Reviewers highlighted several strengths:
- Clear motivation for introducing auxiliary signals;
- Comprehensive consideration of multi-modal inputs and outputs;
- Effectiveness of the memory bank and proposed noise scheduling.

Concerns were raised about novelty, baselines, ablation, and the use of an internal dataset. While most issues were addressed in the rebuttal, novelty concerns remain, as multi-modal generation has been explored in prior works (e.g., VideoJAM), and the segmented noise scheduling extends Diffusion Forcing. Nevertheless, the paper is well-executed and conveys useful insights, and all reviewers retained their borderline-accept scores. The AC considers the merits sufficient to warrant acceptance.